# Development of an Automatic Pipeline for Participation in the CELPP Challenge

**DOI:** 10.3390/ijms23094756

**Published:** 2022-04-26

**Authors:** Marina Miñarro-Lleonar, Sergio Ruiz-Carmona, Daniel Alvarez-Garcia, Peter Schmidtke, Xavier Barril

**Affiliations:** 1Pharmacy Faculty, University of Barcelona, Av. de Joan XXIII 27-31, 08028 Barcelona, Spain; mminarro@ub.edu; 2Baker Heart and Diabetes Institute, Melbourne 3004, Australia; sruizcarmona@gmail.com; 3GAIN Therapeutics, Parc Cientific de Barcelona, Baldiri i Reixac 10, 08029 Barcelona, Spain; dalvarez@gaintherapeutics.com; 4Discngine S.A.S., 79 Avenue Ledru Rollin, 75012 Paris, France; peter.schmidtke@discngine.com; 5Catalan Institute for Research and Advanced Studies (ICREA), Passeig de Lluis Companys 23, 08010 Barcelona, Spain

**Keywords:** docking, D3R, automated pipeline, pocket detection, binding mode prediction

## Abstract

The prediction of how a ligand binds to its target is an essential step for Structure-Based Drug Design (SBDD) methods. Molecular docking is a standard tool to predict the binding mode of a ligand to its macromolecular receptor and to quantify their mutual complementarity, with multiple applications in drug design. However, docking programs do not always find correct solutions, either because they are not sampled or due to inaccuracies in the scoring functions. Quantifying the docking performance in real scenarios is essential to understanding their limitations, managing expectations and guiding future developments. Here, we present a fully automated pipeline for pose prediction validated by participating in the Continuous Evaluation of Ligand Pose Prediction (CELPP) Challenge. Acknowledging the intrinsic limitations of the docking method, we devised a strategy to automatically mine and exploit pre-existing data, defining—whenever possible—empirical restraints to guide the docking process. We prove that the pipeline is able to generate predictions for most of the proposed targets as well as obtain poses with low RMSD values when compared to the crystal structure. All things considered, our pipeline highlights some major challenges in the automatic prediction of protein–ligand complexes, which will be addressed in future versions of the pipeline.

## 1. Introduction

Computational approaches have proven to be a valuable addition to wet-lab techniques in the field of drug discovery [1]. Amongst them, we can find Structure-Based Drug Design (SBDD) methods, where the three-dimensional structure of biomolecules is used to identify small molecules that can interact with them. Predicting how a ligand binds to a target is an essential step for SBDD, and molecular docking has become a standard tool for drug discovery [2,3]. The outcome of docking is a set of proposed positions and conformations of the ligand in the binding site (poses), each with an associated score. These models can be used to interpret and guide ligand design well before the structure of the protein–ligand complex can be experimentally determined.

Nonetheless, docking programs do not always find accurate ligand poses when compared to the experimental solution. There are still challenges that need to be addressed such as receptor flexibility, proper accounting of solvation effects or better scoring functions [3]. Owing to the potential and relevance of docking for SBDD, there has been a substantial and sustained effort to improve the technique, and many docking tools have been developed, such as GLIDE [4], rDock [5], GOLD [6] and AutoDock [7]. Because different docking programs use different sampling strategies and scoring functions, it is important to be able to evaluate and compare the performance between them. To that aim, test sets are available to evaluate the performance of docking and scoring methods in binding mode, binding affinity or virtual screening tasks. Regarding the former application, multiple assessments have been performed with different evaluation benchmarks [8,9,10,11,12,13]. One of the most recent and complete studies was conducted by Wang et al. (2016), who evaluated ten different docking programs, including five commercial programs and five academic programs using a collection of 2002 protein–ligand complexes from the PDB. Concurrently, a strong emphasis has been put on generating highly refined test sets, which only include high-quality structures of relevant protein targets containing drug-like ligands. Some of the most-used validation datasets are CCDC/Astex [14] and Iridium [15]. Such datasets and comparative studies provide a comprehensive understanding of the advantages and limitations of each docking program and help users make more appropriate choices among available methods. However, they suffer from an important limitation: in an attempt to keep the comparison across docking programs fair, the authors of the comparative studies use standard parameters, whereas in real-life applications, advanced users introduce substantial bias to improve performance. In consequence, such comparative studies reveal the intrinsic capabilities of the programs, which is quite different from how they are actually used in typical drug-discovery settings. In addition, as relatively small sets of well-curated protein–ligand complexes become widely adopted as test-sets, there is a risk of biasing docking programs towards those specific datasets.

The challenges organised by the Drug Design Resource (D3R) represent a welcome departure from this tendency. D3R aims to provide benchmark datasets and blinded challenges to assist in the evaluation and improvement of computational algorithms, giving participants the freedom to use the methods as they see fit, but encouraging the use of reproducible protocols. Besides the annual Grand Challenge, D3R also organises the CELPP Challenge (Continuous Evaluation of Ligand Pose Prediction) [16]. Participants in CELPP are encouraged to develop an automated workflow to generate binding mode predictions for different targets that are delivered weekly.

In this article, we describe the development of the first version of a pipeline for participation in the CELPP Challenge, as well as validation results. The main focus of our workflow is to adopt a knowledge-based approach whenever possible, trying to extract data from similar systems that are already deposited in the PDB. Depending on the amount of information available, the docking algorithm may benefit from knowledge about the location of the binding site, specific pharmacophores or even the binding mode of specific substructures. We will describe the different options, analyse their respective performances and identify aspects that need further improvement.

## 2. Results and Discussion

The goal of this work was to create an automated workflow for protein–ligand pose prediction. It must be able to extract information from related complexes deposited in the PDB and to use it in different docking protocols. Throughout this work, a test set consisting of structures released in previous weekly CELPP challenges was used to design the protocol and for benchmarking.

### 2.1. Overview of the Pipeline

One of the key aspects of this work is the automation of the process; therefore, all the steps are gathered in a combination of python, SVL and shell scripts and divided into individually functional modules corresponding to the different phases of the protocol (Figure 1). There are four phases summarized here (see Method section for further details):

*Phase 1: Protein analysis. Download the sequence of the query protein, identify structures of homologous proteins in the PDB and ligands that bind to them (this is performed through a query in 3decision* [17]).


*Phase 2: Ligand analysis. Compute a similarity score and maximum common substructure between the query ligand and all ligands retrieved in Phase 1.*



*Phase 3: Pharmacophore generation. Derive, whenever possible, a pharmacophore for the ligands retrieved in Phase 1.*



*Phase 4: Docking. Three docking strategies are used: tethered docking (when large maximum common substructure (MCS) is shared with a reference ligand), docking with pharmacophoric restraints (if a pharmacophore could be defined in Phase 3) and docking without any restraints (in all cases).*


Additionally, the process includes communication with the CELPP server to download the queries and upload the predictions.

### 2.2. Workflow Input Data, Data Structure and Output

Each weekly CELPP data package is downloaded as a gzipped tar file that contains one directory per target. The target is a protein defined by its primary sequence. Within each directory, there is a set of structures that have the same or highly similar sequences to the target. They are provided as potential receptor structures for docking and contain the highest resolution unbound candidate protein (hiResApo), the highest resolution ligand-bound (hiResHolo), the candidate protein that contains the ligand with the largest MCSS to the target ligand (LMCSS), the candidate protein that contains the ligand with the smallest MCSS (SMCSS) and the candidate protein that contains the ligand with the highest structural similarity (based on Tanimoto score and Daylight fingerprints, as implemented by RDkit [18]) to the target ligand (hiTanimoto). Then, we find the SMILES [19], MOL file and INCHI key [20] corresponding to the target ligand. Finally, the suggested binding pocket centre is also given. However, our pipeline includes a cavity detection phase, so the suggested binding pocket centre will not be used. The expected output from participants is a docked pose of the target ligand with each suggested candidate structure.

### 2.3. Pipeline Development

#### 2.3.1. Blast Results

Before starting the implementation of the pipeline, we analysed the targets from previous CELPP weeks (test set) to check how often they had high similarity homologues already deposited in the RCSB PDB. For this purpose, we ran a blast search against the RCSB PDB with two different identity thresholds: 80% and 95%. From this step, we could conclude that 100% of the targets had some close homolog structure available (>80% identity) within the RCSB PDB prior to its release. When looking for proteins with an identity higher than 95%, we obtained varying results across weeks with an average of 77, 1% of positive cases (Figure 2A). This mirrors the trends in the PDB, which is highly redundant in protein composition [21]. In light of the results, we set the identity threshold for blast searches in our automatic pipeline to 80%.

#### 2.3.2. Ligand Similarity

We analysed the similarity between the ligands provided by CELPP and the ligands obtained by 3decision from similar proteins. After running the 3decision protocol, we were able to obtain sets of ligands for 75% of the proteins in the test set. Using MACSS keys fingerprints, we obtained a mean Tanimoto score of 0.6 with 0.008 and 0.96 being the minimum and the maximum scores obtained, respectively (Figure 2B). We also took into account the size of the compared ligands and their maximum common substructure with a complementary similarity measure, the Tanimoto MCSS [22]. Its value distribution is rather different from the Tanimoto MACSS, (Figure 2C) with average, minimum and maximum values of 0.42, 0.1 and 0.947, respectively.

#### 2.3.3. Docking Method Selection

Using the same target, we compared the performance of the three different docking methods (tethered, pharmacophoric restraints and free) and checked if there was any kind of correlation between the docking RMSD and the Tanimoto similarity to the reference ligands. RMSD values were calculated using the sdrmsd utility from rDock. The mean RSMD values for tethered docking, docking with pharmacophoric restraints and free docking were 2.81 Å, 2.15 Å and 2.19 Å, respectively. Thus, while the use of knowledge-based restraints improved the predictions in individual cases (Figure 3), the overall performance was not better (Table 1). In the case of tethered docking, our analysis showed that it should only be applied when the Tanimoto MCSS is larger than 0.65, after which point almost all predictions were correct (Figure 4A). Unfortunately, this applied to a small proportion of the cases (15%). Surprisingly, free docking also produced improved predictions for this set of ligands, which might be due to the similarity with the ligand of reference used to define the cavity or to the protein pre-organisation (quasi self-docking). The plot also showed that using tethered docking when the MCSS is too small leads to worse predictions than free docking, explaining the apparently worst performance of tethered docking compared to free docking when considering the entire test set. Regarding pharmacophore-guided docking, contrary to our initial expectations, we found that there was not a significant difference in total mean RMSD between restrained and free docking (2.15 Å and 2,19 Å, respectively). This could, in part, be related to the cavity definition process, which already limits the docking space and may leave a small margin for improvement. However, it also suggested that the choice of pharmacophoric restraints was sub-optimal and had to be re-optimised. Thus, we introduced an improved pharmacophore elucidation protocol (see Methods and results below).

#### 2.3.4. Pipeline Effectiveness and Processing Time

The above-described pipeline performance was tested with a collection of pre-released CELPP weeks as well as with the weekly released CELPP set. The execution time of the whole protocol took an average 6.5 min per target. The total execution time varied each week depending on the number of released targets (26 to 68 in the period considered here) and the connection speed to 3decision (from 22 s to 3 min per target). The 3decision protocol could not obtain reference structures for 20% of the targets due to some internal errors of a beta version of the program or because there were no ligands found in druggable pockets from similar proteins. This last event was relatively rare, as it accounted for 25% of times that we were not able to obtain results from 3decision, or 5% of the total. Finally, the similarity analysis to the docked ligand poses took 4.8 min per target on average (Table 2).

### 2.4. Pipeline Validation

To validate the pipeline, we ran it prospectively for a total of 12 weeks. Table 3 shows that the pharmacophoric restrained protocol was the most-used method (51% of the cases). On the other hand, free docking and tethered docking were applied in much lower percentages of cases, 35% and 13.01%, respectively. The mean RMSD value for free docking was 6.2 Å, 5.1 Å for pharmacophore-guided docking and 2.8 Å for tethered docking. However, there is a bigger difference when looking at the proportion of correctly predicted cases by each method. For free docking, only 7.9% of the cases had an RMSD value lower than 2 Å, for pharmacophore guided docking this value increased to 21.4%, and in tethered docking we reached 31.5% of correct poses.

The values obtained with the validation set were much worse than the ones obtained using the test set. The main difference between the sets as that the automatic pipeline for retrieving the cavities using 3decision was not yet automatized during the development stage. In consequence, all the cavities were visually inspected and selected using the 3decision webserver. By contrast, the automatic scripts used at the validation stage to identify the docking cavity and retrieve aligned ligands from 3decision were error-prone. We also had to consider the possibility that the test set was not representative enough of the whole range of systems that can be found in the CELPP Challenge. Nonetheless, the sources of errors and the difference in performance between the test set and validation will be reviewed in the next section.

After analysing the prospective results, we wanted to review if the algorithm for docking protocol selection derived from the test set was the most adequate one. For this purpose, we applied all three protocols to all the validation set and compared the best RMSD obtained for the three methods *(*Figure 4). We could find some differences regarding the accuracy of the docking methods in the test set and validation sets. Tethered docking yielded better results than free docking when MCSS score ≥ 0.5 on the validation set (vs. a marginal improvement on an MCSS score ≥ 0.65 for the test set). Nonetheless, tethered docking was still the method that gave the worst results in low MCSS score values (MCSS < 0.3). As for the pharmacophore-guided docking, during the validation phase, we improved the pharmacophoric elucidation protocol that provided consistently better results than in the test set (see Methods). It also provided improved results compared to free docking in the 0.5 to 1 MCSS score range, with a performance on par with tethered docking. In the 0.25 to 0.5 MCSS score range, pharmacophore-guided docking and free docking performed at a similar level. At lower MCSS score values, free docking outperformed pharmacophore-guided docking.

### 2.5. Challenges to Address

In this section we will describe the most important factors affecting the predictive performance of our pipeline. Figure 5 depicts the main issues and challenges to overcome in the CELPP challenge, which will be treated in more detail in the following sections.

#### 2.5.1. Automated Protocols

When testing a docking program or workflow, a crucial component that will have a big impact in the predictions is the choice of dataset [13]. Usually, the datasets to test docking programs, such as DUD-E [23] or Astex [14], are highly curated datasets, whilst the CELPP receptors are selected automatically and are not manually prepared by experts. Additionally, we have to take into account that CELPP is designed as a cross-docking challenge, which means that we have the added problem of protein flexibility, as the used receptor may not be in the most-fitting position for the ligand. Finally, participants are given, each week, an average of 40 systems to predict and a limited amount of time (3 days), which implies that all the processes need to be automatized, leaving virtually no time for the visual inspection or study of the targets.

In consequence, the pose prediction performance is lower than for other challenges. The median prediction RMSD for the best categories (LMCSS and hiTanimoto receptors) is around 5 Å, being only 20% of the pose predictions accurate within 2 Å [17], whereas reported performance for curated datasets regularly reaches the 80% [13]. Clearly, the latter reflects a best-case scenario, which means that a significant effort to improve automated target structure selection and preparation will be necessary in order to attain better results in CELPP.

#### 2.5.2. Scoring Challenges

Over the past years extensive efforts have been dedicated to improving the existing scoring functions, but nowadays the accuracy of most scoring functions is still a limiting factor in many drug design projects, and results require careful evaluation and post-docking analysis.

To assess the accuracy of the docking score, we selected a subset of 446 submitted cases and checked if the submitted pose is the one with the lowest RMSD compared to the crystal structure. In 208 out of 446 total cases (46.6%) the docking protocol was able to produce a correct pose (RMSD lower than 2 Å), but in 75 of them, the pose with the lowest RMSD was not ranked as the best solution by rDock’s intermolecular score (SCORE.INTER). This translates to a 64% success rate when the correct pose can be generated. Note that this is close to the 76% success rate obtained on the CCDC-Astex Diverse Set, a standard test set for binding mode prediction where correct predictions can be generated for 99% of cases [5].

Figure 6 shows the median RMSD obtained with the different receptors for the submitted pose and for the best pose generated by the pipeline. The median RMSD for the submitted pose was around 4.18 Å, whereas if we considered the best prediction, the mean decreased to 2.9 Å and the median to 2.4 Å. From these results, it is evident that the pipeline would benefit greatly from a complementary method to re-score the docking poses. An approach that presented better results in other blind challenges [24] was the combination of the docking scores with Dynamic Undocking (DUck) [25,26] simulations of the top-scoring poses. By combining both methods, we expected to be able to obtain a more accurate pose ranking for challenge submission.

#### 2.5.3. Sampling Challenges

##### Cavity Selection

The CELPP Challenge is designed as a pose prediction challenge and to assess the influence of receptor choice in docking performance. For that reason, the coordinates for the centre of the cavity are provided by the organisers. Nonetheless, we wanted to go one step further by creating a pipeline of general applicability and add a cavity selection step to our protocol, thus avoiding the need to pre-define the binding site. The cavity detection is performed automatically by 3decision, and all the possible cavities are retrieved and considered for docking. The method that 3decision uses for cavity detection is fpocket, a pocket detection algorithm based on Voronoi tessellation [27]. When more than one cavity is detected, our pipeline selects the cavity based on the similarity of the ligands retrieved by 3decision with the target ligand. On average, 3.2 cavities were detected per target, but in 67 cases (14%), the correct cavity was not detected, and so the docking was carried out in the wrong cavity. Figure 7 shows an example where 3decision only detected the cavity represented by the grey surface, missing the actual cavity represented by the green surface. In 9% of cases, the failure corresponded to shallow cavities on the protein surface that are not detected by the fpocket algorithm.

Another reason for not detecting the cavity correctly (14% of cases) is that the ligands bind at the interface of a dimer, but only one protein is reported in the challenge. Note that, unlike other docking challenges or scenarios, the receptors provided by CELPP are not manually curated. They rely on a fully Automated Pipeline to perform that task, which can sometimes lead to the selection of inappropriate structures (e.g., giving a monomer instead of a dimer) for obtaining an accurate ligand pose [17]. Figure 8A shows one such example. The remaining failures in this category were attributed to an error with the API when downloading the analysis results.

##### Docking Method Selection

In our protocol we implemented three different docking strategies that were applied depending on the different set thresholds. From the 305 cases of the validation set where we did not obtain the correct pose, in 78 cases the correct binding pose had been correctly predicted by a different docking strategy.

As shown in Table 4, from those 78 cases, only in 9 cases the correct solution was found by free docking instead of a form of guided docking. By contrast, 26 cases could have been correctly predicted if a form of guided docking had been used instead of free docking. This analysis also reveals that the two forms of guided docking employed here are not equivalent: 27 incorrect pharmacophore-guided docking solutions were correctly predicted by tethered docking. Vice versa, 16 incorrect tethered docking solutions were correctly predicted by pharmacophore-guided docking. One such example is shown in Figure 9. These results suggest that all the binding poses generated by the different docking protocols should be considered, then rescored with a post-docking method to identify the best one [28].

##### Receptor Flexibility

As pointed out by many previous studies [29], receptor flexibility is an important factor that can alter docking predictions. Both small changes on side-chain orientation and bigger structural changes can lead to incorrect predictions [30]. We could attest to this phenomenon when docking against the different proposed receptors. For each target, the docking protocol was run using all the receptors provided by the organisers. Figure 6 displays the validation results categorised by the receptor. The best-performing receptor was LMCSS, which corresponds to the one hosting the ligand most similar to the query. SMCSS obtained the worst results, with a median RMSD of 5.9 Å.

As an example, Figure 10 shows two cases where the differences in side-chain orientation of residues from the binding site are interfering with the correct binding position. In the case of 6pl1 (Figure 10A), there is a difference in the conformation of a loop in the binding site of all the receptors used that cause Phe 669 (in blue) to block part of the binding site obtaining a totally different cavity. It is established that, by using a variety of receptor conformations, we increased the probability of generating a correct ligand pose, but selecting the optimal docking cavity remains a major challenge for docking methods [31,32]. This result also highlights the need to select multiple binding mode predictions, which should be re-scored with a more rigorous computational methodology.

##### Other Molecules in the Binding Site

This pipeline was intended for general applicability, and for this reason, during the cavity preparation process all the ligands and co-solvents were removed, and only the coordinates of the receptor were kept. However, in some systems, especially enzymes, cofactors can have an important role in determining the ligand binding mode. Two such examples are provided in Figure 11. Lastly, the fact that there can be other molecules in the binding site can interfere when generating the pharmacophoric restraints. As they are in the same cavity, our protocol included them in the list of retrieved ligands from similar proteins, and those are considered in the pharmacophoric restraint generation pipeline.

## 3. Materials and Methods

### 3.1. Candidate Preparation

For each candidate structure, co-crystallized solvent and ligands were removed using Schrödinger’s split structure tool [34], and only the coordinates of the receptor were kept. Subsequently, the protein preparation tool from MOE [35] was used to fix problems within the crystal structure, and the Protonate 3D tool [36] was used to assign protonation states to the protein (assuming pH 7.0). All the files were saved in Tripos MOL2 format, as required by the docking program, rDock [5]. All the above steps were integrated in an SVL script for automation purposes.

### 3.2. Ligand Preparation

We took the query ligand in SMILES string format and used the LigPrep tool from Schrödinger [37] to calculate the 3D structure with the proper topology; tautomerism; bond orders and geometry of bonds, angles, dihedrals and rings. Additionally, the ionizable groups were protonated at pH 7 with a threshold of ±1 pH unit. All ligands were saved in SDF format.

### 3.3. Selection of Similar Proteins, Druggable Pockets and Ligand Retrieval

One of the pillars of the whole process was being able to select good reference systems from which we could extract some restraints to guide our docking predictions. For this purpose, we integrated into the pipeline a protocol based on the *3decision* tool from Discngine; 3decision [17] is a web-based platform that centralizes all structural knowledge (including all the RCSB PDB dataset) to perform multiple kinds of analyses. We queried 3decision using a dedicated REST API endpoint. Using as input the target sequence in FASTA format, a blast against the database was performed to select those proteins that share a high identity (I% > 80%). The 3decision database also contains all pre-computed druggable pockets as predicted by the fpocket cavity detection tool [27]. The pockets are aligned based on the sequence and superimposed to the query structure. Finally, we exported all the ligands found in the aligned pockets in an SDF file, which was also converted to SMILES format using Openbabel [38]. In the case where multiple druggable pockets were detected, the corresponding docking protocol was applied to every pocket.

### 3.4. Ligand Similarity and Maximum Common Substructure Calculation

After retrieving the ligands found in similar pockets, a similarity analysis was performed between the query ligand and the list of retrieved ligands using MACCS keys fingerprints and the Tanimoto coefficient scoring, which has been identified as one of best metrics for similarity calculations [39]. The Tanimoto coefficients as well as the fingerprints were calculated using rdkit [19].

The maximum common substructure (MCSS) between the target ligand and the ligands retrieved from similar proteins was calculated using RDKit’s FindMCS function [19]. As a complementary measure of similarity between the ligands, and also working as a method to evaluate the robustness of the MCSS, a Tanimoto coefficient based on MCSS was calculated using Equation (1) [22].
(1)$$Tanimoto_{MCSS}=\frac{N_{AB}}{\left(N_{A}+N_{B}\right)-N_{AB}}$$
where $N_{A}$ and $N_{B}$ are the number of heavy atoms in molecules *A* and *B*, respectively, and $N_{AB}$ is the number of heavy atoms in the MCSS. The $Tanimoto_{MCSS}$ can have values between 0 and 1, 1 being the value obtained when two molecules are identical.

### 3.5. Generation of Pharmacophoric Restraints

Ligand-based pharmacophore modelling has had a great impact in drug discovery [40]. In this work, this strategy was used to extract common chemical features from the aligned ligands retrieved by 3decision before elucidating the pharmacophores. The Align-it tool from Silicos-it [41] was used to generate a combination of pharmacophore points for each molecule in the set. In this work two different versions of the protocol for the generation of a consensus pharmacophore were tested. In the first version, after the generation of the pharmacophoric points for each molecule, the features that were common between molecules were selected and ranked by number of appearances, and then the two highest ranked features were selected and used as mandatory pharmacophoric restraints for docking. In the second Version, the ligands were first clustered based on similarity (MACCS fingerprints and Tanimoto similarity of 0.9). From each cluster, the ligand corresponding to the centroid was selected, thus removing redundancy and obtaining a diverse set of ligands, and then the pharmacophoric points were generated. From here, only the most-representative points (those shared by more than 45% of the ligands) were considered as mandatory restraints. Points shared by between 20% and 44% of the ligands were considered optional restraints. For the optional restraints, at least one of them needed to be fulfilled during the docking process.

### 3.6. Molecular Docking

To perform all the docking processes, we used rDock [5], a fast, versatile and open-source docking program. To run rDock, we needed the prepared receptor structure and a definition of the binding site. To define the binding site in this work, we chose the reference ligand method with rDock’s default parameters. From the pool of retrieved ligands, we selected as a reference ligand the one having the maximum sum of the MACCS Tanimoto similarity score and $Tanimoto_{MCSS}$ score. This combined score implies a similar ligand and also a similar size to the target ligand. As a result, the cavity size was adapted to the query ligand, adding another restriction level to the docking process.

After ligand preparation, rDock is able to explore exocyclic bond rotations on the fly using a genetic algorithm together with rotations and translations. Conveniently, rDock can perform free docking as well as different types of restraint docking. Using rDock capabilities, our pipeline could use three different docking protocols, depending on the characteristics of the system and the available information. If we found a good reference ligand ($Tanimoto_{MCSS}\;$ > 0.5), then the pipeline would choose tethered docking, fixing the MCSS with the *sdtether* utility. Otherwise, if there was a sufficient number of diverse ligands to extract a pharmacophore (>5), a pharmacophoric restraint docking was chosen instead. Finally, unrestrained docking was used for the remaining cases. All the docking predictions used the standard rDock docking protocol (*dock.prm*).

### 3.7. Pose Selection

The output from the pipeline was a set of poses generated by the docking program for each candidate structure in an SDF file. Then, the poses were sorted by rDock’s intermolecular score (SCORE.INTER), which accounts for the protein–ligand interaction’s free energy. Formally, solutions should be sorted based on the total score, which accounts for the intramolecular energy as well (SCORE.INTRA + SCORE.INTER), but it has been shown that the intramolecular term bears a large error and can introduce more noise than signal to the predictions [42]. Using *sdsort,* the best pose was selected and saved in an SDF file. If more than one cavity was detected, this selection protocol was then applied to each cavity. Thereafter, the cavities were ranked based on the MCSS score obtained during the Ligand similarity and MCSS calculation, and then the best poses from each cavity were ranked by rDock’s SCORE.INTER. The best scoring pose from the top scoring pocket was then selected for submission. Finally, the files were transformed to the format required by CELPP submission rules: the ligand pose in MOL format and the receptor in PDB format.

## 4. Conclusions

Quantifying the performance of docking software in real scenarios is essential to understanding their limitations, managing expectations and guiding future developments. With the CELPP Challenge, D3R aimed to provide a fast-growing validation set that better captures all the complexity in a real drug-discovery setting. Here we presented an initial version of our pipeline for participation on the CELPP Challenge, which applies different knowledge-based docking approaches depending on the already available information on PDB.

To provide a baseline performance, the CELPP team developed four workflows based on different docking programs, one being rDock. The rDock workflow represents a default implementation of the method without any optimisation and using the cavity defined by the challenge. Our protocol had the added challenge of detecting the cavity automatically, but when we considered only the cases where the cavity was correctly predicted, we observed a significant performance of our protocol relative to the baseline, with improvements in the median RMSD value ranging from 1.0 Å to 2.6 Å, depending on the docking cavity (Figure 6). This confirms that gathering information from already-deposited complexes in PDB and transforming them into the appropriate restraints benefits the docking process greatly.

Our final goal was to evolve this platform into a docking server where more rigorous, but also more computationally demanding methods, could be applied (e.g., molecular dynamics). Nonetheless, there are some additional points that need to be revised. The first one is cavity detection and characterization. For our pipeline being able to identify possible binding sites for the majority of targets, 3decision has proven to be a valuable tool. However, there are some cases where the 3decision protocol is not able to retrieve the correct pocket because they are shallow cavities or the receptor structure is ill-defined. In this first version of the pipeline, targets where there is no pocket information are neglected, and no docking protocol is applied. For these situations, we could use a local implementation of fpocket [38] to check whether there are, in fact, no possible druggable cavities. Another option would be using molecular dynamics with co-solvent/water mixtures (MDmix) [43,44] to identify possible binding sites. Nonetheless, we would like to add the option of taking the cavity coordinates as a reference. With this, we would separate the cavity-finding problem from the docking problem, reduce execution time and increase the predictive power when the binding site is already known.

A second point to revisit is the choice of receptor structure. As discussed, protein flexibility is an important aspect to consider in a drug-discovery setup. Proteins can adapt their structures to the bound ligand, so using an apo structure or one in a complex with a very different compound degrades the performance of the docking program. One way to mitigate this effect would be to use different conformations of the receptor and select the one with the better score as the optimal structure [45].

A third aspect is the management of ‘third-party’ molecules in the binding site, namely cofactors and water molecules. In this initial version of the pipeline, all systems are processed and prepared in the same way, stripping the binding site of all non-protein molecules. However, we detected several cases where docking failed owing to missing cofactor molecules that should be considered part of the receptor. This can be solved with a curated list of cofactors that should not be removed. Water molecules are frequently found at the protein–ligand interface, mediating hydrogen bonds between the partners. By keeping these structural waters on the binding site, the ligand pose predictions can be more accurate.

We will also continue to monitor the performance of restrained and unrestrained docking in prospective CELPP predictions. As previously shown, by using the MCSS score, we are able to determine which is the docking method that performs best for each case. Initially, we applied a rather restrictive cutoff of 0.65, which included only 13% of the total cases. After considering all the participation cases, we were able to determine better ranges of applications for each type of docking protocol, which presently is set to 0.5 and includes 31% of cases.

As far as the creation of the pharmacophores, in cases where, due to a lack of pre-existing information when ligand-based pharmacophore cannot be extracted, we could make use of hot spots derived from the structure. Such hot spots can be identified by their ability to bind small organic co-solvents [43,46]. By performing molecular dynamics with co-solvent/water mixtures (MDmix), we can identify binding sites and hot spots [47] that could be used as pharmacophoric restraints for docking. The addition of this methodology to our workflow would also allow us to assess the druggability of the pockets selected by 3decision.

## Figures and Tables

**Figure 1 ijms-23-04756-f001:**
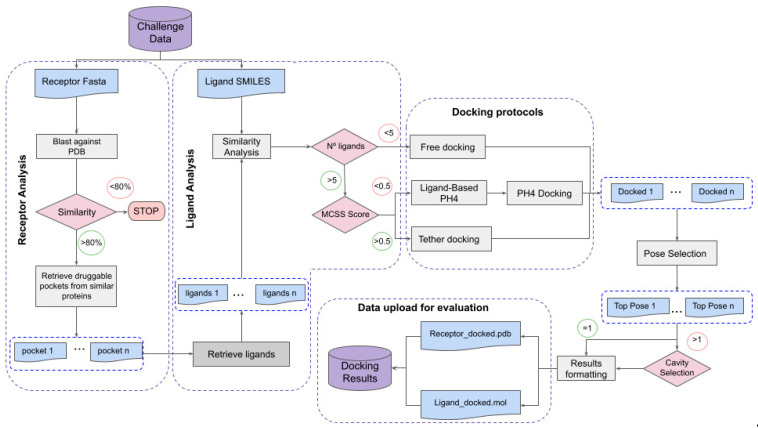
Workflow employed for pose prediction.

**Figure 2 ijms-23-04756-f002:**
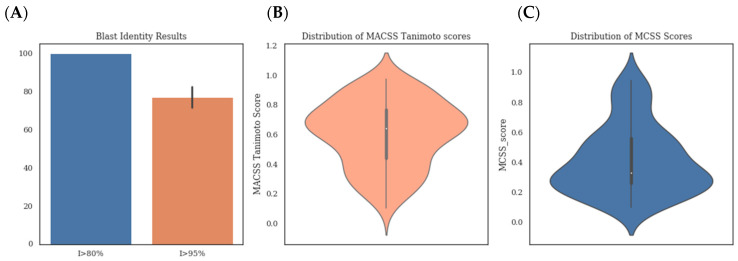
(**A**) Histogram representing the percentage of targets for which we obtained blast results with and identity higher than 80% (blue) and 95% (marron) (**B**) Distribution of Tanimoto MACSS score and (**C**) Tanimoto MCSS scores obtained for the ligands in the test set.

**Figure 3 ijms-23-04756-f003:**
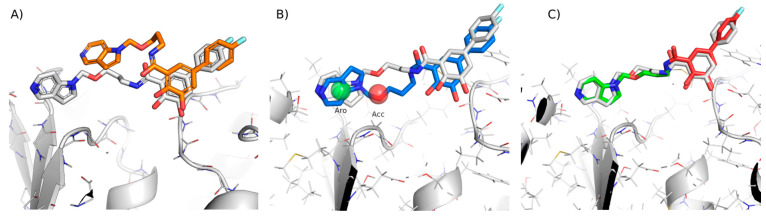
Differences in best pose predicted for target 5p8y from CELPP week 33. Image (**A**) corresponds to free docking with an RMSD of 4.09 Å. Image (**B**) is the best prediction obtained with pharmacophoric restraints (1.74 Å). Image (**C**) corresponds to the best pose using tethered docking, obtaining an RMSD of 0.95 Å. The red substructure indicates the tethered atoms.

**Figure 4 ijms-23-04756-f004:**
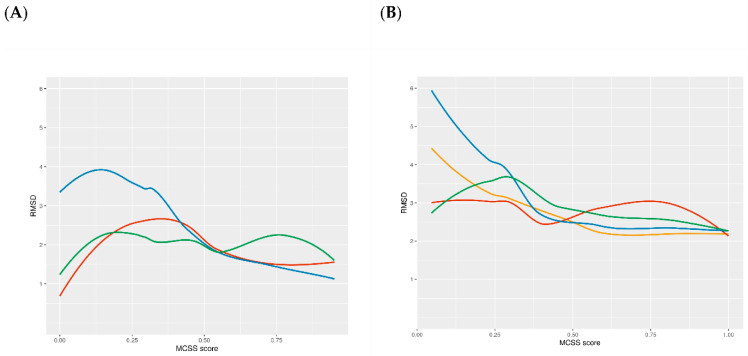
Relation between RMSD and the MCSS score using (**A**) the test set and (**B**) the validation set. Free docking results shown in red, docking with pharmacophoric restraints in green (version 1) and yellow (version 2; only applied to the validation set) and MCS-tethered docking in blue.

**Figure 5 ijms-23-04756-f005:**
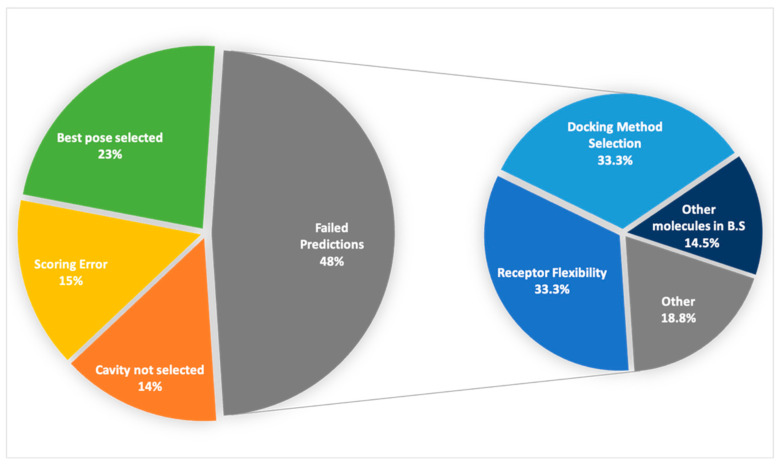
Overall view of validation set cases.

**Figure 6 ijms-23-04756-f006:**
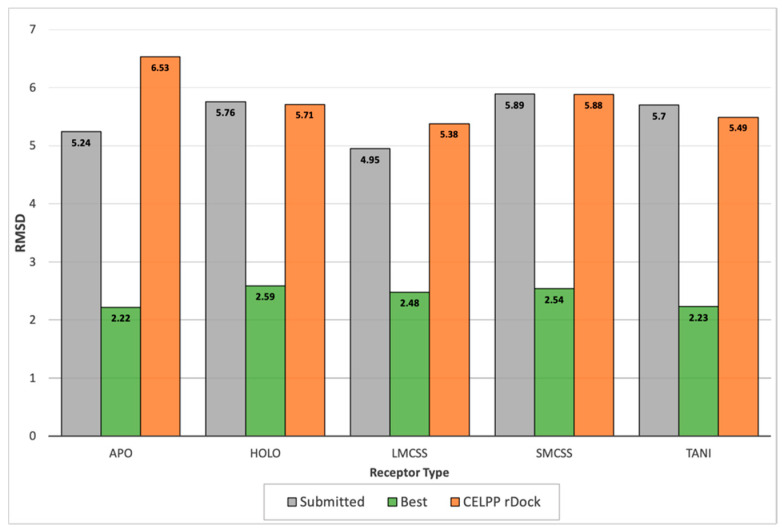
Median RMSD for the submitted pose compared to the best pose generated by the pipeline. The CELPP rDock workflow values are obtained from the D3R website (https://drugdesigndata.org/about/celpp2-charts accessed on 1 May 2021).

**Figure 7 ijms-23-04756-f007:**
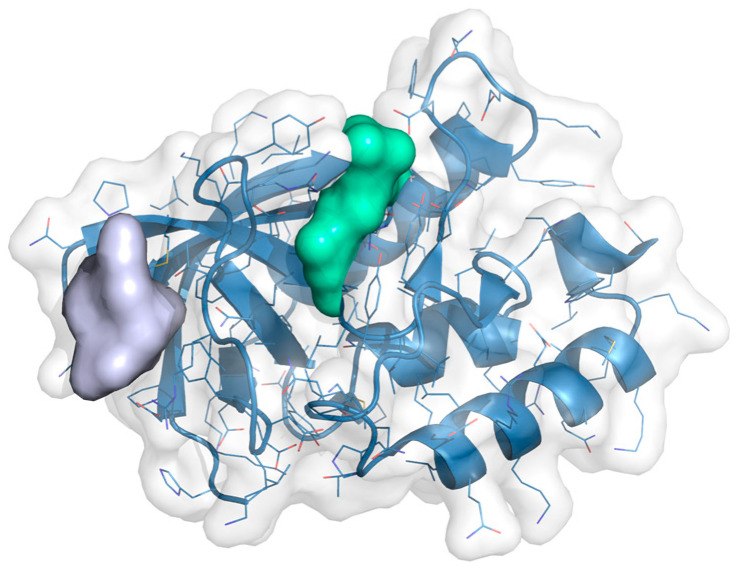
PDB 6ok9 with the pocket detected by 3decision represented by the purple surface and the correct pocket represented by the green surface.

**Figure 8 ijms-23-04756-f008:**
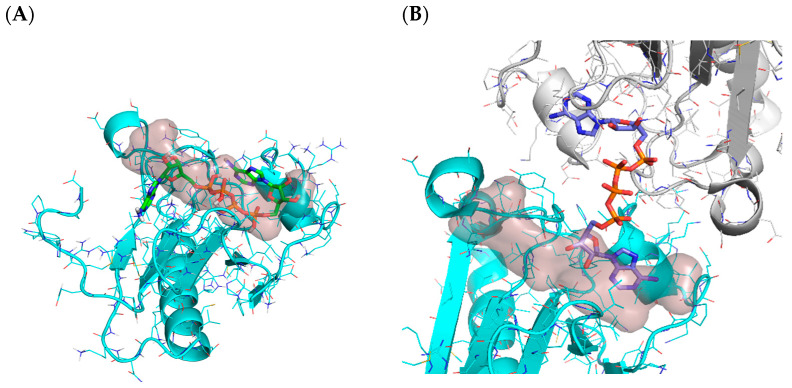
(**A**) hiTanimoto receptor for target 6j65. The solution selected by the pipeline is represented as sticks. (**B**) Protein dimer in PDB code 6j65. The crystalized ligand is represented as sticks. For both figures, the reference cavity provided by 3decision is shown as transparent surface.

**Figure 9 ijms-23-04756-f009:**
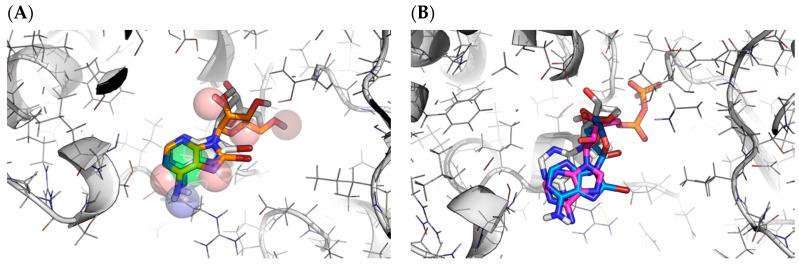
Predictions for PDB 6dfo and hiTanimoto Receptor using: (**A**) Pharmacophoric restraints. Predicted pose in orange. Pharmacophore represented as spheres. (**B**) Tether docking. Predicted pose in blue. Reference ligand in pink. In both cases, the crystallographic solution is shown in white for reference. The RMSD values with the predicted poses are 1.2 Å and 3.3 Å, respectively.

**Figure 10 ijms-23-04756-f010:**
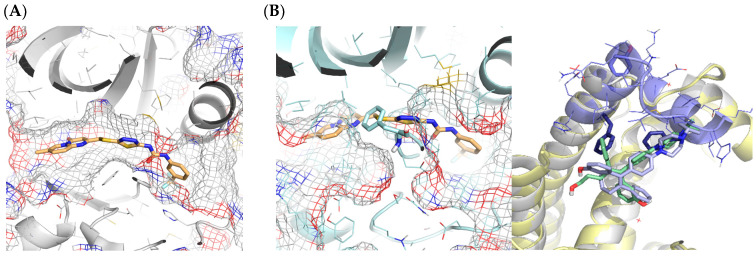
(**A**) Differences in binding site structure organisation between 6pl1 crystal and the selected hiTanimoto receptor by CELPP; the correct ligand pose is represented in beige, (**B**) Differences in site conformations for target 6a6k between receptor hiResHolo in purple, the crystal structure in white and hiTanimoto receptor in yellow. The ligand crystal pose is represented in green and in light purple is the pose obtained using the hiResHolo receptor.

**Figure 11 ijms-23-04756-f011:**
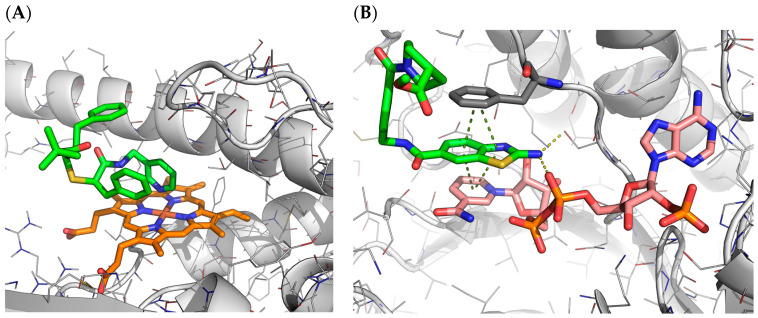
(**A**) Interaction of ligand G0D (green) with heme group (orange) in PDB 6DA2 [33].Ligand belongs to a series of analogues with pyridine as a heme-ligating head that works as an inhibitor of CYP3A4 by decreasing the heme reduction rates [33]. (**B**) Interaction of ligand EV8 (green) and NADP (pink) in PDB 6gd0. In yellow dashed lines are H-bond interactions and in green dashed lines π interactions.

**Table 1 ijms-23-04756-t001:** RMSD results obtained using different docking methods.

	Free Docking	Tethered Docking	Ph4 Docking
Mean	2.19	2.81	2.15
Median	1.96	1.71	1.63
Min	0.43	0.33	0.39
max	7.41	15.07	7.41

RMSD values in Å.

**Table 2 ijms-23-04756-t002:** Statistics of the pipeline implementation CELPP weeks.

	No. of Targets	3decision Time	Docking Time	Total Time
Week1	31	34	103	137
Week2	44	103	174	277
Week3	27	10	113	123
Week4	43	118	176	294
Week5	29	35	153	188
Week6	40	182	265	447
Week7	68	234	123	357
Week8	26	102	111	213
Week9	28	126	247	373
Week10	48	158	382	540
Week11	50	193	270	463
Week12	26	137	716	853
Mean	38	119.33	236.08	355.42

Time measured in minutes.

**Table 3 ijms-23-04756-t003:** RMSD values and percentage of cases for each docking protocol.

	Free Docking	Ph4 Docking	Tethered Docking
Mean	6.2	5.1	2.8
Std	6.2	3.4	1.6
Min	1	0.5	0.7
Q1	3.9	2.2	1.6
Q2	6.3	4.7	2.3
Q3	8.2	7.7	3.6
max	13.6	13.9	12.7
<=2Å	7.9%	51%	13%
Application rate	35%	51%	13%

RMSD values in Å.

**Table 4 ijms-23-04756-t004:** Comparison between the submitted docking method vs. the method that yields the best result.

		Best Prediction		
		Free	Ph4	Tethered
Submitted	Free		6	20
	Ph4	8		27
	Tethered	1	16	

## Data Availability

All the systems mentioned in this paper were provided by the CELPP organizers. Data regarding the CELPP challenge can be found in https://drugdesigndata.org/about/celpp2-charts (accessed on 31 March 2022).

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
