# Peer review of "Development of an Automatic Pipeline for Participation in the CELPP Challenge"

_ijms, 2022, doi:10.3390/ijms23094756_

Round 1

Reviewer 1 Report

The authors nicely presented  an automated tool for predicting the ligands pose based on the existing knowledge and by performing the docking. The RMSD scores are reasonably well and the authors do mentioned the difficulties involved for correct pose prediction. I strongly recommend the manuscript to be accepted with minor revision. Here are few comments from my side:

It would be nice to report which particular targets fails to report lower RMSD scores in order to identify the difficulty involved in there. Knowing about them tells the difficulty about the target and scope for better prediction. 

How the ligand flexibility is handled in the docking calculations? Or is it rigid?

Is there any way to report binding affinity/scores as well along with pose prediction? This will give additional information to be compared with the exp. binding affinity and in rank-ordering of the ligands based on the binding affinity/scores. 

Is this tool publicly available to use ? An example for the public to reproduce/try would be great. 

Author Response

Dear Reviewer,

Thank you for your positive feedback and interest in our work. In response to our manuscript, you raise some really interesting questions that we hope we can properly address in this letter. 

1- It would be nice to report which particular targets fails to report lower RMSD scores in order to identify the difficulty involved in there. Knowing about them tells the difficulty about the target and scope for better prediction.

Firstly, the idea of finding a way of grouping the most challenging targets (i.e by protein family) and studying in detail how to approach them is in fact really interesting.We have explored this possibility but we have encountered some difficulties. Unfortunately, there is a large variability in the results and not sufficient number of ligands per target to make a statistically meaningful comparison of the performance on a specific target family.  We have observed that this large variability in the results is mainly due to the fact that the information generated in previous CELPP weeks is used to guide the docking in more recent weeks, so the performance for a particular family can vary from week to week. 

Having said that, although we are not able to answer this question with the data we have at the moment, we think that the thorough analysis of the different docking challenges across protein families is a really relevant subject that deserves its own independent study. 

2-How the ligand flexibility is handled in the docking calculations? Or is it rigid?

Regarding ligand flexibility, rDock explores exocyclic bond rotations on the fly (torsions are explored by the genetic algorithm together with rotations and translations). Ring conformations are pre-generated with other software (in this case LigPrep) and are treated as a rigid body by rDock. We have clarified this point in the Methods Section (page 14 line 483).

3-Is there any way to report binding affinity/scores as well along with pose prediction? This will give additional information to be compared with the exp. binding affinity and in rank-ordering of the ligands based on the binding affinity/scores.

Although having the binding affinities of the compounds would be an interesting addition to the process, the CELPP challenge is purely a docking pose challenge where only the ability to recreate the crystal pose is assessed. Maybe in the future, we could add different functionalities to our protocol to participate in other challenges involving affinity prediction like the Grand Challenge organized by D3R, which indeed evaluates pose-prediction, affinity rankings, and also free energy calculations. 

4-Is this tool publicly available to use ? An example for the public to reproduce/try would be great.

At the moment the tool is not publicly available, as it makes use of some commercial software not free for distribution. Our future goal is to adapt this platform into a docking server publicly available.

Reviewer 2 Report

The authors have presented an automated pipeline for pose prediction validated by participating in the Continuous Evaluation of Ligand Pose Prediction (CELPP) Challenge. The findings are very interesting, and method, analysis, and conclusions appear reasonable and justified. I recommend publication after a minor revision:

1) The legends A) or a) and B) or b) in figures 4, 9 and 11 must be homogenized.

2) In Figure 5, the colours of the "failed predictions" graph should be changed to make it more readable.

3) In Figure 6, the colours of the data series "Best" or "CELPP rDOCK" must be changed in order to be more readable.

4) In the legend of Figure 7 it is mentioned a grey surface is not purple?

5) In equation (1) a smaller font size must be used.

6) In section 2.3.2. Ligand Similarity, line 143 the bibliographic reference is missing.

7) In section 2.5.3. Sampling Challenges, line 294 the bibliographic reference is missing.

8) Some typos need to be corrected, namely:

- section 2.3.3. Docking method selection, line 155;

- section 2.5.1. Automated protocols, line 244;

- section 3.7. Generation of Pharmacophoric Restraints, line 448.

Author Response

Dear Reviewer,

Thank you for your positive feedback and interest in our work. We applied all of your suggestions in the new version of the manuscript. First, all the image legends have been homogenized to lowercase. Then, the color of the plots depicted in figures 5 and 6 have been changed to avoid any misunderstanding in the interpretation of the results. In the legend of figure 7, we changed the surface color from gray to purple so it matches the image better. Also, the font size for equation 1 has been reduced. Finally, all the missing references have been added (sections 2.3.2 and 2.5.2) and the spelling mistakes have been also corrected (sections 2.3.3, 2.5.1 and 3.7).